# Stability-Enhanced Emission Based on Biophotonic Crystals in Liquid Crystal Random Lasers

**DOI:** 10.3390/ma16010200

**Published:** 2022-12-26

**Authors:** Zhenzhen Shang, Zhi Wang, Guang Dai

**Affiliations:** 1Tropical Biodiversity and Bioresource Utilization Laboratory, Qiongtai Normal University, Haikou 571127, China; 2School of Physics, Beijing Institute of Technology, Beijing 100081, China; 3College of Science, Tianjin University of Technology, Tianjin 300384, China

**Keywords:** liquid crystal, random laser, biophotonic crystals, butterfly wing, multiple scattering

## Abstract

A new design of a bio-random laser based on a butterfly wing structure and ITO glass is proposed in this article. Firstly, the butterfly wing structure was integrated in a liquid crystal cell made of ITO glass. The integrated liquid crystal cell was injected with liquid crystal and dye to obtain a bio-random laser. A non-biological random laser was obtained with a capillary glass tube, liquid crystal and dye. The excitation spectra and thresholds were recorded to evaluate the performance of the biological and non-biological random lasers. The results show that the excitation performance stability of the bio-random laser is improved and the number of spikes in the spectra is reduced compared with the non-biological random laser. Finally, the equivalent cavity length of the biological and non-biological random lasers was compared and the optical field distribution inside the butterfly wing structure was analyzed. The data show that the improvement of the excitation performance stability of the bio-random laser is related to the localization of the optical field induced by the photonic crystal structure in the butterfly wing.

## 1. Introduction

With the rapid development in the fields of nano-integration, biosensing, and optical imaging, there is a growing demand for miniature, flexible, and high-quality optoelectronic devices. Random lasers, as nonconventional lasers, are a promising light source the feedback mechanism of which relies on the scattering elements embedded in the active materials rather than the well-defined optical cavities [1]. Furthermore, random lasers have potential applications in photonic marking, sensing and speckle-free imaging due to their broad angular emission, low spatial coherence and high-brightness [1,2]. Most importantly, as a fixed resonant cavity is not required, random lasers have the characteristics of a small size, flexible shape and low cost, which can meet the current requirements for its application as an optoelectronic device [3,4].

Nevertheless, due to the random distribution of the scattered particles in a random laser system, the lasing spectrum is varied and unstable. In order to control the random laser emission, scientists have utilized a variety of materials to design random lasers, including semiconductor [5], liquid crystal [6,7], metal nanostructure [8,9], thin film [10], wrinkled graphene oxides [11], waveguide structure [12,13] and even perovskite single crystal [14]. 

Biomaterials, because of their biodegradability, good compatibility and flexibility are currently attracting extensive attention from random laser researchers. Random lasing from diverse biomaterials has been investigated, such as with human tissue [15,16], silk protein [17,18], ordinary paper [19] and so on. In the fabrication of random laser devices, biomaterials can be used not only as scattering particles, but also as gain media and are easier to stretch and recover [20,21,22]. Insect wings, such as butterflies and cicadas, are relatively common biological materials. It is well known that photonic crystals are artificial microstructures made of periodic arrangements of media with different refractive indices [23,24,25]. However, natural photonic crystal microstructures exist within the butterfly wing structure [26], which can yield unique optical phenomena, such as optical interference, diffraction and scattering effects [27,28,29].

In fact, Chen et al. [29] has observed random laser emission from the wing structure of a butterfly coated with zinc oxide powder. In a subsequent study, Zhou et al. also conducted a random laser experiment with butterfly wings and observed a coherent random laser [27].

They prepared a bio-random laser covered with magnesium fluoride, which is unidirectional. Recently, they achieved speckle-free imaging of random lasers produced by moth wings [30]. Nevertheless, they only designed random lasers based on dye and butterfly wing. Moreover, there is no report on whether the butterfly wing structure has the function of modulating random lasers.

In this paper, we will adopt ITO glass to make a bi-directional (two-sided) light emitting random laser. We also attempt to design random lasers based on liquid crystal. After testing, it was found that the butterfly wing could be used to achieve a stability-improved random laser by simply integrating the butterfly wing structures into the liquid crystal random lasers. The role of the butterfly wing structure in improving the stability of the random laser emission pattern is confirmed by comparing the spectra from the samples with and without butterfly wings. In addition, the butterfly wing structure can also modulate the number of random laser emission patterns.

## 2. Materials and Methods

Materials: Solutions of Pyrromethene-597 (PM597) in acetone at 10^−2^ M and 4-(dicyanomethylene)-2-methyl-6-(p-dimethylaminostyryl)-4H-pyran (DCM, Exciton Inc., Dayton, OH, USA) in ethanol at 10^−3^ M were adopted to provide the gain. A nematic liquid crystal (P06161A, produced by Shijiazhuang Cheng-zhi Yong-hua Display Materials Co., Ltd., Shijiazhuang, China) with a refractive index $n_{e}=1.72$, Δn=0.19 was used at room temperature (~300 K) without any treatment. ITO conductive glass (2 cm by 2 cm) and capillary glass (diameter: 5 mm, length: 50 mm) were employed in the sample preparation. The butterfly wings (Pieris rapae crucivora) were captured near Dong Shang village, Fei Cheng city, Shan Dong province, the panoramic image and SEM image of which are shown in Figure 1a. Figure 1d shows the fluorescence spectrum of the dye DCM.

Samples preparation: Table 1 shows the samples and their components used in the experiment. First, the wings were soaked in ethanol for approximately 8 h to remove impurities and scales from the wing surface, which were subsequently placed on the glass and rinsed repeatedly with deionized water. Next, the washed wings were dried in a dust-free environment. Finally, one part of the dried wing was cut into 1 cm pieces with scissors, and the other part was ground into powder (about 10 microns). The method for preparing each sample using the above materials is as follows. (1) DCM/LC: Mixed 0.2 mL ethanol solution containing DCM and 1 mL liquid crystal thoroughly, then used a capillary glass tube to draw part of the solution to make a sample. (2) DCM/LC/Wing: The above butterfly wings (1 cm) were fixed between two glass sheets, and then the above mixed DCM/LC solution was dropped on the unsealed part of the glass sheet, and the solution penetrated the sample by capillary action (the sketch of DCM/LC sample preparation is shown in Figure 1c). The thickness of the gap between the two glass sheets is approximately 20 microns. (3) PM597/Wing: The butterfly wing powder was added to the acetone dispersion of PM597 (1 mL) and, after stirring uniformly, the dispersion was extracted through the capillary action of a capillary glass tube.

Experimental setup: The experimental setup of the optical measurement is shown in Figure 2. A second-harmonic of a Q-switched Nd: YAG laser with a 532 nm wavelength, 10 Hz repetition rate and 8 ns pulse duration was used in the experiment. A polarizer and a half-wave plate are placed behind the laser to adjust the pump intensity. A beam splitter then splits the laser beam into two parts, one of which is the reference beam and the other is used to excite the sample. The reference beam is collected by an energy meter and the excitation beam is perpendicular to the sample after passing through a half-wave plate and a cylindrical lens. The cylindrical lens eventually converges the pump light into a strip light source, 9 mm in length and 0.16 mm in width. Finally, we used a spectrometer (with resolution 0.13 nm and fiber aperture 400 microns) connected to a computer to collect the excitation spectra on the two outer surfaces and sides of the sample. The single-shot spectrum, recorded by the spectrometer, is obtained from 10 pump pulses. The non-single-shot spectra in this paper are the ensemble-averaged emission spectra for a sum of the single-shot spectrum (sum = 20).

## 3. Results and Discussion

The spectra shown in Figure 3a,b are the emission of the DCM/LC and the DCM/LC/Wing, respectively. In Figure 3a, only the broad photoluminescence with a full-width at half-maximum (FWHM) of 25 nm can be observed at a pumping energy of 27.9 μJ, which is viewed as the consequence of an amplified spontaneous emission (ASE), meaning that the gain of the system does not compensate for the loss. When the pumping energy increased to 49.5 μJ, the FWHM of the emission spectrum narrowed to approximately 8 nm. As we increased the pump energy further, for example, to 69.9 μJ, a number of peaks with FWHMs of less than 0.3 nm were observed, indicating the presence of a random laser. For a gain-scattering system, gain-related excited radiation can trigger light amplification; multiple scattering effects associated with disordered scattering particles cause the system to display threshold behavior. When the external pump energy is greater than the threshold, the gain of the system is greater than the loss and, in turn, triggers random laser. That is, the random laser behavior from the DCM/LC is related to the multiple scattering induced by the randomly distributed liquid crystal molecules inside the sample (see Figure 4).

In fact, the wavelength position of the spikes in Figure 3a varies with the pumping energy. This indicates that, although the liquid crystal molecules provide sufficient scattering feedback for the random lasing action to overcome the loss, the randomness of their distribution still leads to the instability of the spectrum [31]. Similar to Figure 3a, the emission FWHM from the DCM/LC/Wing decreases with pumping, from 0.65 nm at 168 μJ to 0.3 nm at 183 μJ (Figure 3b), where the spectral narrowing above the threshold is the characteristic of lasing. Figure 3c,d shows the emission intensity and FWHMs (select the highest spikes for measurement) as a function of pumping energy for the DCM/LC and DCM/LC/Wing, respectively. The threshold behavior in Figure 3c,d further confirms the random laser behavior in the DCM/LC and DCM/LC/Wing samples.

It is clear that the number of spikes from the DCM/LC/Wing emission spectra (Figure 3b) is significantly reduced compared to that of DCM/LC (Figure 3a). At the same pumping energy of 234 μJ, the number of spikes reduced from 18 to 8 (see the inset in Figure 3d). As it can be seen in Figure 1a, the multiple window-like microstructures with a near-periodic distribution are shown in the butterfly wing structure. Therefore, the random laser spectra emitted from the DCM/LC/Wing sample are related to both the randomly distributed liquid crystal molecules within the sample and the microstructures on the butterfly wings.

Next, we measured the spectra from three other detection locations of the DCM/LC/Wing (Figure 5). The spectra acquired in Figure 5a–c, presented the same characteristic as in Figure 3b, with randomly distributed sharp peaks on top that show constant wavelength positions independent of the pumping energy. This shows that the microstructures of the butterfly wing enhance the localization of the random cavity, and compared with the other modes, the local mode is dominant in the random laser emission. The emission intensity and FWHM, as a function of pumping energy from the DCM/LC/Wing at the three other detection locations, are plotted in Figure 5d–f. Significant threshold behavior was observed, corresponding to the threshold values of 61.3, 33.9, and 80.3 μJ, respectively. For visual comparison, the thresholds for the four different detection locations of the DCM/LC/Wing are shown in Figure 6a. Compared with DCM/LC, the threshold of DCM/LC/Wing either increased or decreased at four different pump positions (see Figure 6a). The higher threshold of 161.3 μJ is attributed to the lower reflectance of the butterfly wing surface, which reduces the energy inside the cavity and increases the scattering feedback loss [2].

We extracted the data from Figure 3b and Figure 5 into Figure 6a,b, where the solid blue line depicts many isolated and compact points (position of the peaks). It can be seen that the emission spectra of the DCM/LC/Wing have fewer isolated peaks, implying that the butterfly wings influence the reduction in the number of random laser emission modes. For the microcavity of the lasing, the resonant frequency is [32]:(1)$$\omega_{0}=\frac{2\delta \omega_{0}\int \varepsilon_{r}{\left|E\left(r_{i}\right)\right|}^{2}dV}{-\alpha_{\mathrm{ex}}{\left|E\left(r_{i}\right)\right|}^{2}}$$
where $\varepsilon_{r}$ is the dielectric constant of the medium inside the cavity; $\alpha_{\mathrm{ex}}$ denotes the polarization intensity of the molecules in the cavity; $\delta \omega_{0}$ represents the magnitude of the resonant wavelength shift after filling the cavity with molecules; $E\left(r_{i}\right)$ is the electric field at the location of the molecule, which is related to the adhesion position of the molecule. In Figure 6b, the peak positions of the emission spectra are basically unchanged at different pumping positions. These stable resonance wavelengths indicate that the liquid crystal molecules are uniformly distributed in the wing structure.

Figure 7a,b shows the single shot spectra of the DCM/LC and DCM/LC/Wing, respectively, recorded when the pump energy and pump position are fixed (recorded every 3 min). The location of the spikes marked by arrows in Figure 7a are plotted in Figure 7c. It can be seen that the spike positions of the DCM/LC emission spectra at different moments are unstable. However, the spike positions of the random laser spectra from DCM/LC/Wing at various moments, shown in Figure 7b, are almost immobile, which reaffirms the modulation effect of the butterfly wing microstructure on the random laser.

Next, we performed a power Fourier Transform (PFT) analysis of the emission spectra (inset of Figure 3d), as given in Figure 7d. For a random laser, the length of the equivalent resonant cavity is defined as [33]:(2)$$L=d_{1}\pi /n$$
where n is the refraction index of the gain medium (1.7) and $d_{1}$ is the first peak in the PFT curve. The cavity length of the DCM/LC/Wing is 2.0 μm, which is smaller than that of DCM/LC (4.8 μm). That is, the travel path of the photons in the DCM/LC/Wing system is significantly shorter than that in the DCM/LC system, which means that the microstructures of the butterfly wing play a local role in the process of light propagation. In fact, the cavity length size of the DCM/LC/Wing is approximately equal to the individual microcavity length observed in the SEM image (marked with green lines in Figure 1a). This implies that the microcavity in the butterfly wing structure does play a role in the stability regulation of the random laser. As shown in Figure 1b, we used FDTD time-threshold finite-difference simulations to analyze the light field distribution inside the butterfly wing structure when visible light shines on it. Because the window-like microstructures inside the butterfly wing structure are periodically distributed, we only draw the light field distribution within one periodic structure, where the refractive index of the butterfly wing structure is 1.56. It is evident that the light field is mainly localized near the edge of the butterfly wing structure. This again confirms the role of localization in the microstructures of butterfly wings, which is consistent with our discussion in Figure 5.

To confirm the modulation effect of the butterfly wing structure on the random laser, we subsequently prepared a random laser containing only the butterfly wing structure. The variation curve of the PM597/Wing emission spectrum with pumping energy is given in Figure 8a. When the pump energy is 123 μJ, the PM597/Wing only emits an ASE spectrum with a FWHM of 25 nm. As the pump energy increases to 161 μJ, a narrow wave peak with a FWHM of 1.5 nm appears in the emission spectrum. Further increasing the pumping power to 244 μJ, a narrower crest suddenly appears, corresponding to a FWHM of approximately 0.3 nm, revealing the occurrence of lasing action. However, when pumping a sample containing only dye PM597, the sample emitted only ASE as the pump energy increased to 477 µJ (see the inset in Figure 8d). Therefore, it is determined that that the peaks from thePM597/Wing are related to the role of the butterfly wing structures. The multiple scattering caused by the butterfly wing powder inside the capillary glass tube provides the feedback needed for the generation of the random laser. The output intensity of the PM597/Wing sample and its emission linewidth are plotted in Figure 8b as a function of pumping energy. The output intensity increases with the pumping energy and the lasing threshold is found at ~161 μJ, reconfirming the random laser emission behavior.

As a function of the pumping energy, as shown in Figure 8a, the amplitude of each peak increases while its resonant wavelength remains almost unaltered. This again illustrates the enhancement of the random laser performance stability by the optical localization induced by the photonic crystal structure inside the butterfly wings. To confirm this conjecture, the single shot spectra (record every 3 min) from PM597/Wing are presented in Figure 8c at a fixed pumping position and energy, where the resonance wavelengths remain stable. In Figure 8d, we measured the mode intervals of the two adjacent spikes, based on the five spikes marked in Figure 8c, corresponding to the values of 1.08, 0.98, 1.02 and 1.07 nm, respectively. These stable data reconfirm the role of the butterfly wing microstructure in regulating the random laser stability.

## 4. Conclusions

In summary, a method of designing a biological random laser using ITO glass and butterfly wings based on the existing research is presented in this paper. The adoption of ITO glass is essential for future attempts to modulate bio-random lasers using an electric field. We demonstrate the role of biological microstructures in butterfly wings in improving the stability of random lasers. Firstly, we integrated the butterfly wing structure into the liquid crystal random laser (DCM/LC) and obtained the DCM/LC/Wing random laser. Next, by comparing the excitation spectra of the two random lasers, DCM/LC and DCM/LC/Wing, it is found that the butterfly wing structure significantly enhances the original excitation stability of the DCM/LC random laser. Meanwhile, by analyzing the resonant cavity lengths of the two random lasers and the light field distribution inside the butterfly wings, the role of the butterfly wing structure in enhancing the localization of the light field was determined. Finally, we prepared a random laser containing only a butterfly wing structure and gain medium, and its excitation data again confirmed the role of butterfly wing structure in regulating the stability of random laser.

## Figures and Tables

**Figure 1 materials-16-00200-f001:**
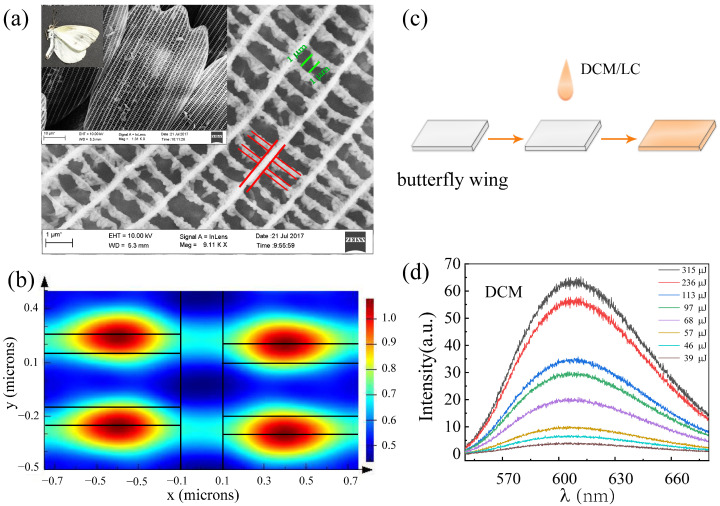
(**a**) Panoramic image and SEM images of butterfly samples (Pieris rapae crucivora); (**b**) The light field distribution inside the structure of the butterfly when visible light is shone on its wings (the black lines in the figure correspond to the red lines marked in (**a**)); (**c**) Preparation process of DCM/LC/Wing samples; (**d**) Fluorescence spectra of dye DCM at different pump energies.

**Figure 2 materials-16-00200-f002:**
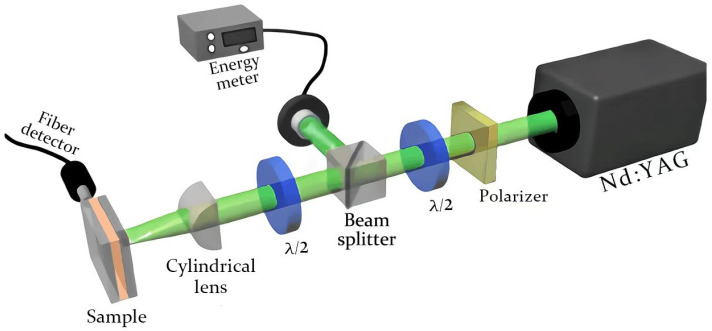
The experimental setup of optical measurement.

**Figure 3 materials-16-00200-f003:**
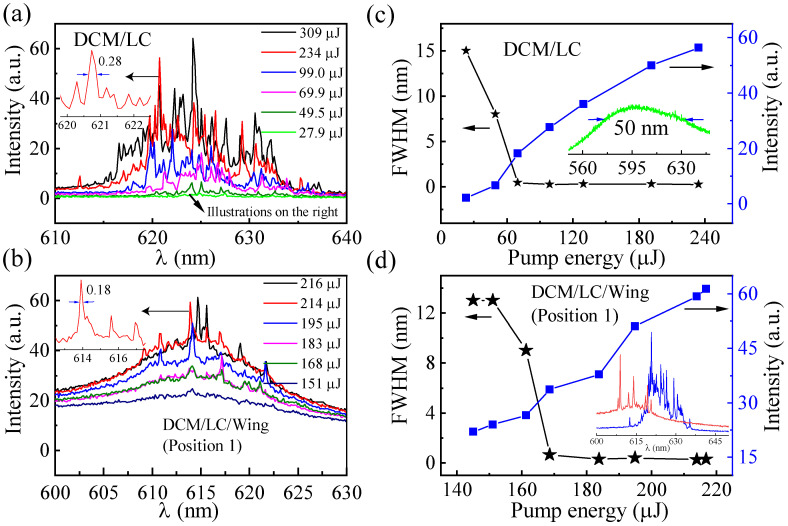
Emission spectra as a function of pumping energy for (**a**) DCM/LC and (**b**) DCM/LC/Wing; The emission intensity (blue squares) and full-width at half-maximum (black stars) as a function of the pumping energy for (**c**) DCM/LC and (**d**) DCM/LC/Wing. Illustration in (**c**): Enlargement of the green line in (**a**). Illustration in (**d**): The evolution of emission spectra recorded for DCM/LC and DCM/LC/Wing at the pump energy of 234 μJ.

**Figure 4 materials-16-00200-f004:**
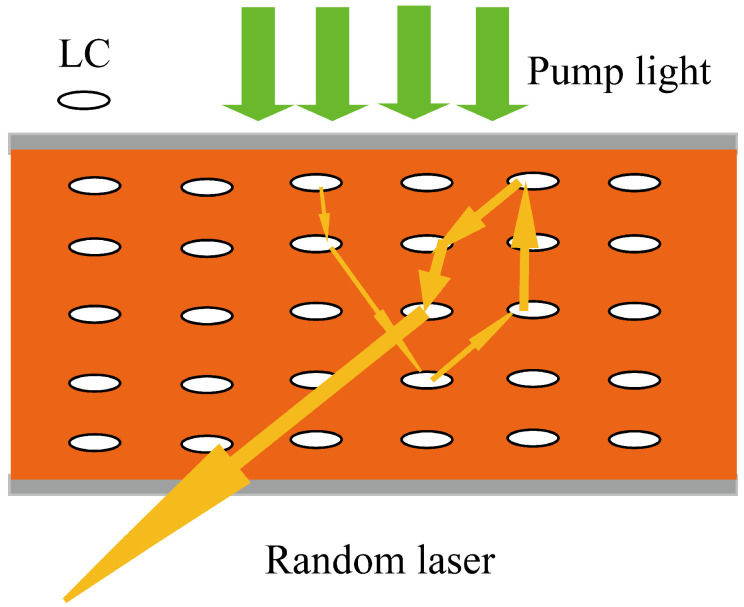
Multiple scattering of light caused by liquid crystal molecules in the gaining medium, green arrows: pump light, white circles: liquid crystal molecules, yellow arrows: path of light.

**Figure 5 materials-16-00200-f005:**
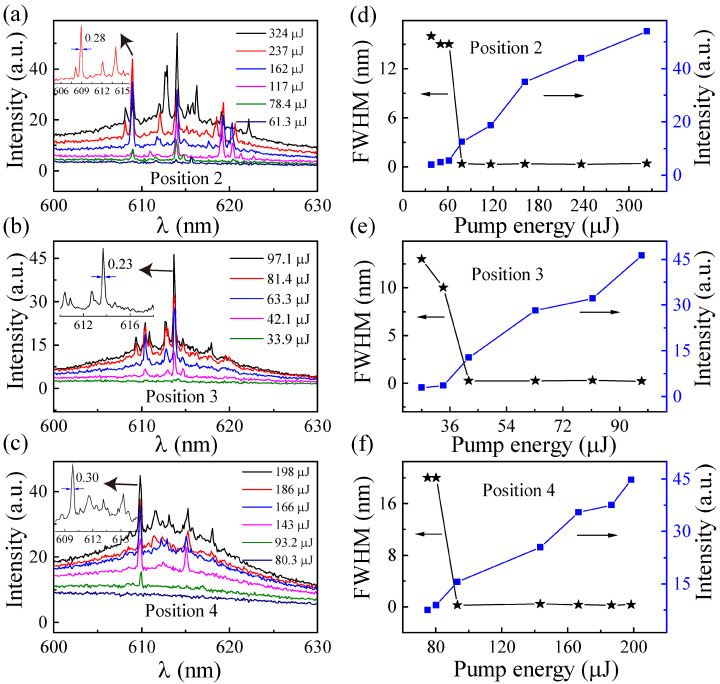
(**a**–**c**) Emission spectra as a function of pumping energy with different pumping positions for DCM/LC/Wing; (**d**–**f**) The emission intensity (blue squares) and FWHM (black stars) as a function of the pumping energy corresponding to (**a**–**c**).

**Figure 6 materials-16-00200-f006:**
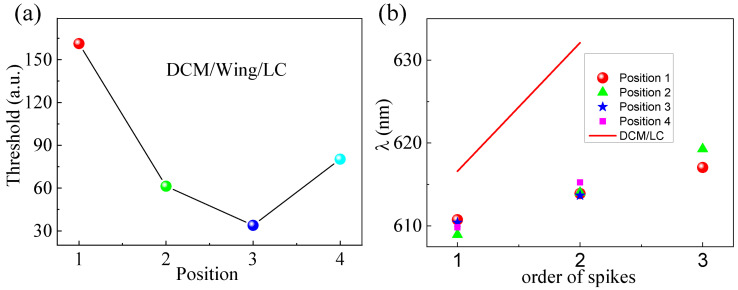
(**a**) Laser threshold of DCM/LC/Wing under different pumped positions, four colors represent four different pumped positions; (**b**) Peaks positions extracted from Figure 3a,b and Figure 5a–c.

**Figure 7 materials-16-00200-f007:**
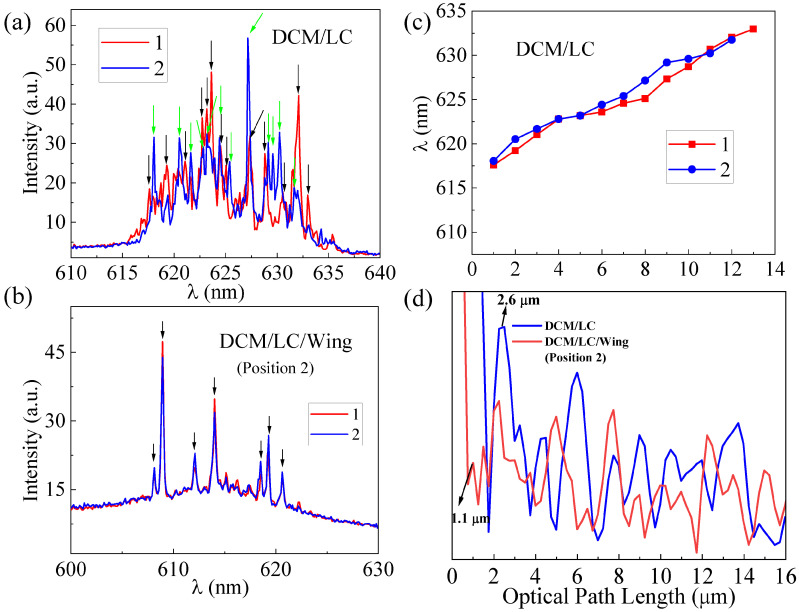
(**a**) Single shot spectra of DCM/LC at the pumping energy of 309 μJ, green and black arrows: the peak positions of the two single shot spectra; (**b**) Single shot spectra of DCM/LC/Wing at Position 2 with the pumping energy of 260 μJ; (**c**) The location of the spikes marked by arrows in (**a**); (**d**) Power Fourier Transform analysis for the emission spectra in the inset of Figure 3d.

**Figure 8 materials-16-00200-f008:**
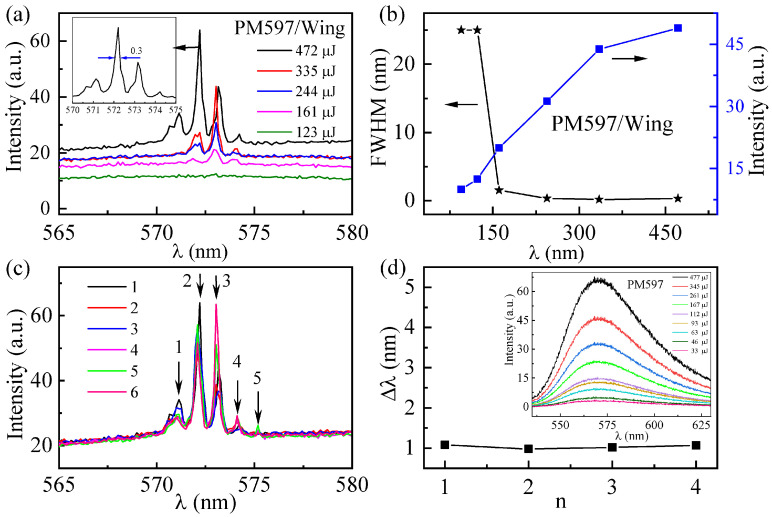
(**a**) Emission spectra of PM597/Wing under different pumping energies; (**b**) The emission intensity (blue squares) and FWHM (black stars) as a function of the pumping energy for PM597/Wing; (**c**) Single shot spectra of PM597/Wing with fixed pumping position and pumping energy (595 μJ), the arrow indicates the peak; (**d**) the interval between adjacent patterns from the marked peaks in (**c**). Inset: Fluorescence spectrum of dye PM597.

**Table 1 materials-16-00200-t001:** The samples and their components used in the experiment.

Sample	Type	DCM (mL)	PM597 (mL)	Liquid Crystal (mL)	Capillary Glass (5 × 50 mm)	Glass Substrate (2 × 2 cm)
DCM/LC	-	0.2	-	1	1	-
DCM/LC/Wing	Wing	0.2	-	1	-	1
PM597/Wing	Powder		1	-	-	-

## Data Availability

Data sharing not applicable.

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
