# Peer review of "Stability-Enhanced Emission Based on Biophotonic Crystals in Liquid Crystal Random Lasers"

_materials, 2022, doi:10.3390/ma16010200_

Round 1
Reviewer 1 Report
The authors report on the stability improvement of a random laser due to the introduction of a butterfly wing. Unfortunately, I cannot recommend the work for publication – the claims are not justified and organized.
Line 10 – I am not sure low coherence is a good laser property.
I slightly disagree with the statement in the introduction that random lasers can be a cheaper version of conventional lasers. Applications of those 2 classes are very different and performances incomparable.
Fig 1 is not discussed – simulations – why the bars are displaced, what are the refractive indexes, why a plane wave was chosen for the excitation ets.
Abbreviations, DCM etc are not justified
According to fig 3, the wing makes things worth – the threshold is 2-3 times higher with the wind
The main claim of the paper is the lasing stability – I didn’t find where it was accurately analyzed and compared in the case without a wing.
What is the real role of the wing? Just a reflector – according to the simulation, the index is rather small, so the optical impact is also minor.
Reviewer 2 Report
Questions:
1. On line 44-45, authors have mentioned that Zhou et al. have published work on random lasing effect in butterfly wings. So, in that case what is the novelty of the work presented here? How is this work different than Zhou et al. (citation 24). Also, if Zhou et al. has already published the work on random lasing, then does the first line of conclusion (line 205) needs any modification? Please clarify.
2. Why have authors not explored different concentration of butterfly wings to explore further the enhanced lasing due to wing structure?
3. Authors have not discussed the wing structure that is demonstrated in SEM image in Figure 1a? How the does the cavity structure in the SEM image correlate to the results obtained from photoluminescence experiments?
4. In the experimental setup, the authors have used polarizer and half wave plate to adjust the pump intensity (as claimed by them in line 89-90). Why is a neutral density filter not used? Have authors studied the effect of polarization on the random lasing from LC/DCM/Wings?
5. In the experimental setup, the energy meter is connected at the beam splitter to measure the energy incident on the sample. However, are the energy losses from other half wave plate and cylindrical lens taken into consideration?
6. Also, why was continuous wave laser not used for pumping energy? What heating of the LC the concern? Does the spectrophotometer measure the instantaneous spectra, or is the data acquired a time-averaged spectra since you have used pulsed laser with 10Hz frequency?
7. In Figure 3b and d, authors have not mentioned which peak they have chosen to measure the FWHM? Also, the scattering effect from the alignment of the LC needs to be discussed in detail. For the lasing effect, the alignment of wing structure is very important. How does the wings align with LC. Is it randomly dispersed or is it aligned with the LC?
8. Authors need to explain the details of the position used to study the lasing effect. How is light incident on the LC cell? It is unclear what positions are used.
Reviewer 3 Report
The manuscript entitled “ Stability-enhanced Emission Based on Biophotonic Crystals in 2 Liquid Crystal Random Lasers” by Zhenzhen Shang and Guang Dai describes the experimental studies of the fluorescence and its power dependence in a composite structure based on the DCM dye, liquid crystals and butterfly wings. The main finding is the speckle-like spectrum of fluorescence with the narrow spectral peaks, which are attributed by the authors to lasing effect. They also show that the presence of the butterfly wings modifies the FL parameters of the composite material.
While the idea of this research is attractive, the paper is not quite clear for understanding, and the presented results are not well justified.
Comments to the manuscript:
- The experimental set-up does not contain the parameters of the irradiated area – the laser spot diameter. It seems necessary to provide the information about the size of the homogeneous areas in the butterfly wings. What does it mean that “we collect emission spectra around the sample” (line 93)? Please describe. As well as the angle of incidence and collection aperture.
- Please explain what are the FWHM data plotted in Fig. 3 b,d – do these data correspond to the whole spectrum or to single spectral (speckle) peaks? This is important to clear up as this is the main sign of lasing. The same for Fig. 4.
- Why the average width of the spectra of DCM and of DCM/LC differ so much? Please explain.
- In the introductory part of the manuscript it is announced that the butterfly wings behave as photonic crystals, what about the studied samples? In p. 137, Fabry-Perot resonance is announced – what are the evidences for this statement? It should be confirmed by model estimates, the mode numbers should be estimated, too, and compared with the experiment. The graph in Fig. 5 d does not look as an equidistant FP spectrum.
- Please explain how were the data shown in Fig. 1b obtained and prove that the studied butterfly wings behave as photonic crystals.
Other comments:
- The procedure of the preparation of the butterfly wings is perhaps more suitable for the subsection “samples preparation”.
- Figure 3, a does not allow to distinguish all the data, e.g. to estimate the FWHM discussed in the text. It would be also more convenient to use the same scales in Fig. 3a and Fig. 3c in order to compare them.
- There are a number of typos in the text:
p. 3, l. 88: “HZ” - Hz
p.3, l.90 : “A beam splitter then splits the laser” – perhaps it splits the laser beam?
Reviewer 4 Report
The review report of the manuscript has been attached herewith.

Round 2
Reviewer 3 Report
The revised version of the manuscript “ Stability-enhanced Emission Based on Biophotonic Crystals in Liquid Crystal Random Lasers” by Zhenzhen Shang et al. was substantially modified by the authors. They also underlined the role of the ITO film in the operation of the considered random lasing structure.
Still there are some more comments:
- In the abstract – please explain in a more clear way what is “spectra of two types of random lasers at different moments”
- Also in the abstract : “stable excitation modulation of the random laser by the local action of the butterfly wing microstructure.” - this sentence is unclear as well.
- What is the size of the LC molecules?
- In Fig. 3a, it is absolutely impossible to see the spectral line at 27.9 μJ excitation energy and see whether there are no narrow peaks in it.
- It seems not quite correct to plot FWHM (in Fig. 3) of the average emission spectra (which remains under high pump energies, too) with that of narrow peaks at high energies. It looks as comparison of different quantitaties. For the case of speckle spectrum, how FWHM is estimated? Does it correspond to a single peak, or it is the average characteristics?
- Please explain why the lasing threshold in the case of DCM/LC/Wing is so much higher as compared to DCM/LC structure? Does it indicate that the properties of the random laser are improved as the butterfly wing is added (Figure 3)?
- One reads “threshold sizes were 161.3, 61.3, 33.9, and 80.3 μJ” – what is the accuracy of estimation of these values?
- It is not discussed in the text what is the mechanism of the influence of the butterfly wings in changing the ASE spectra, especially in regulating the stability of the structure. What are the microcavities and their effect announced in the sentence “feedback formed between microcavities in butterfly wings” ? Is that photonic crystals, or what?
Language ^
- “ have potential applications in display applications” (in introduction)
- Too many “materials” in: “In the fabrication of random laser devices, biomaterials can be used not only as scattering materials, but also as luminescent materials [20,21]. Moreover, unlike synthetic stretchable materials [22], biological materials are easier to stretch and recover.”
- Caption to Fig. 1d – it seems better to indicate the difference between the spectra shown in this panel.
- “Materials” section: “.. were prepared for sample preparation..”
- “a near-periodic distribution are distributed”
- In Eq. 2, n means the refractive index. What about Fig. 6 b (X-axis)?
- Conclusions: “In summary, this paper prepared a bio-random laser” – it looks that the paper can not prepare anything itself.
Reviewer 4 Report
The manuscript can be accepted in its present form.
